# MRI-Based Risk Factors for Adverse Maternal Outcomes in Prophylactic Aortic Balloon Occlusion for Placenta Accreta Spectrum and Placenta Previa

**DOI:** 10.3390/diagnostics14030333

**Published:** 2024-02-04

**Authors:** Hiroyuki Tokue, Masashi Ebara, Takayuki Yokota, Hiroyuki Yasui, Azusa Tokue, Yoshito Tsushima

**Affiliations:** Department of Diagnostic and Interventional Radiology, Gunma University Hospital, 3-39-22 Showa-Machi, Maebashi 371-8511, Gunma, Japan

**Keywords:** prophylactic aortic balloon occlusion, placenta accreta spectrum, placenta previa, magnetic resonance imaging

## Abstract

Purpose: We previously reported that T2 dark bands and placental bulges observed in magnetic resonance imaging (MRI) can predict adverse maternal outcomes in patients with placenta accreta spectrum (PAS) and placenta previa undergoing prophylactic balloon occlusion of the internal iliac artery. On the other hand, the risk factors associated with the use of prophylactic aortic balloon occlusion (PABO) have not been sufficiently investigated. This retrospective study aimed to identify MRI-based risk factors associated with adverse maternal outcomes in the context of PABO during a cesarean section (CS) for PAS and placenta previa. Materials and Methods: Ethical approval was obtained for a data analysis of 40 patients diagnosed with PAS and placenta previa undergoing PABO during a CS. Clinical records, MRI features, and procedural details were examined. The inclusion criteria for the massive bleeding group were as follows: an estimated blood loss (EBL) > 2500 mL, packed red blood cell (pRBC) transfusion (>4 units), and the need for a hysterectomy or transcatheter arterial embolization after delivery. The massive and nonmassive bleeding groups were compared. Results: Among the 22 patients, those in the massive bleeding group showed significantly longer operative durations, a higher EBL (*p* < 0.001), an increased number of pRBC transfusions (*p* < 0.001), and prolonged postoperative hospital stays (*p* < 0.05). T2 dark bands on MRI were significant predictors of adverse outcomes (*p* < 0.05). Conclusion: T2 dark bands on MRI were crucial predictors of adverse maternal outcomes in patients undergoing PABO for PAS or placenta previa during a CS. Recognizing these MRI features proactively indicates the need for effective management strategies during childbirth and emphasizes the importance of further prospective studies to validate and enhance these findings.

## 1. Introduction

The global landscape of obstetric care is witnessing a paradigm shift marked by rising cesarean section (CS) rates, particularly in high-income countries. The World Health Organization has reported that global CS rates have significantly increased from approximately 7% in 1990 to 21% in 2022, surpassing the ideal acceptable CS rate, which is approximately 10–15%. This surge in CSs contributes substantially to the increased incidence of placenta accreta spectrum (PAS) and placenta previa worldwide, presenting a shared challenge for maternal–fetal health [1].

PAS refers to abnormal attachment of the placenta to the uterine wall. Increased rates of CSs contribute to changes in the uterine tissue [2]. This condition poses a significant challenge to maternal–fetal health. Placenta previa occurs when the placenta partially or completely covers the cervix, potentially causing bleeding during pregnancy and complications during childbirth [2]. PAS occurs in 5–10% of placenta previa cases and is more likely to occur after CS [2].

PAS and placenta previa are complex obstetric complications characterized by abnormal placental implantation, often resulting in severe maternal morbidity and mortality [2,3]. Despite advancements in obstetric care, managing these conditions remains a formidable challenge, necessitating innovative approaches to enhance patient outcomes [4]. Previous studies have explored the potential benefits of prophylactic aortic balloon occlusion (PABO) in mitigating hemorrhagic complications during cesarean delivery for PAS and placenta previa [3]. Specific risk factors contributing to patient outcomes in PABO remain poorly understood.

Magnetic resonance imaging (MRI) has been established as a valuable diagnostic modality that enables detailed assessment of placental abnormalities [5]. The existing literature underscores the utility of MRI in providing insights into the extent and characteristics of abnormal placental implantation [5,6].

We previously reported that T2 dark bands and placental bulges observed in MRI can predict adverse maternal outcomes in patients with PAS and placenta previa undergoing prophylactic balloon occlusion of the internal iliac artery [7]. However, the specific MRI-based risk factors that influence outcomes in patients with PAS and placenta previa following PABO are yet to be elucidated. The intricate interplay between MRI markers and clinical outcomes remains a significant gap in the current knowledge. Understanding these unknown factors is imperative to refine patient management strategies and improve overall obstetric care.

This study aimed to systematically identify and analyze MRI-based risk factors associated with patient outcomes in the management of PABO for PAS and placenta previa.

## 2. Materials and Methods

### 2.1. Study Design

This retrospective study received ethical approval from our hospital’s Ethics Committee. Due to the retrospective nature of the study, the requirement for written informed consent was waived. We conducted a retrospective analysis of the clinical records of 40 consecutive patients diagnosed with PAS and placenta previa. These patients underwent PABO during a cesarean section (CS) between January 2006 and December 2023. Before CSs, all patients underwent abdominal MRI at our department. In this study, the term PAS includes all three grades of abnormal trophoblastic invasion (placenta increta, percreta, and accreta). CSs were scheduled, and the decision to perform a hysterectomy was made based on intraoperative findings. In all cases, the objective was to preserve the uterus as much as possible through placental removal, uterine reconstruction, and the use of PABO.

### 2.2. Inclusion and Exclusion Criteria

The inclusion criteria were defined as follows: a diagnosis of PAS and placenta previa established through MRI and confirmed intraoperatively, no occurrence of hemorrhage before surgery, gestational age exceeding 28 weeks, access to the patient’s history of CS, presence of placenta previa with the placenta covering the prior cesarean scar, preoperative hemoglobin levels surpassing 10.0 g/L, PABO conducted for all hemodynamically stable patients, and a singleton pregnancy. Exclusion criteria comprised significant fetal motion artifacts on MRI, severe obstetric complications, the administration of coagulation-affecting drugs, fetal anomalies, fetal growth restriction, or planned hysterectomy.

### 2.3. Clinical Data Collection

The clinical records were thoroughly examined for possible risk factors contributing to adverse maternal outcomes, including maternal age, parity, number of prior CSs, and degree of placental adhesion. Additional recorded parameters included gestational age at examination, fetal weight, operative duration, estimated blood loss (EBL), hysterectomy, number of transfused packed red blood cells (pRBCs), and length of postoperative hospital stay. The massive bleeding group was defined according to specific criteria, including an EBL > 2500 mL, pRBC transfusion > 4 units, or the need for hysterectomy or transcatheter arterial embolization after delivery. The nonmassive bleeding group had an EBL of <2500 mL and uterine preservation [8].

In cases of total hysterectomy, the uterine and placental tissues were examined histologically to strengthen the surgical diagnosis in cases of suspected deep invasion.

### 2.4. MRI Protocols

Placental MRI was performed on all patients using a 1.5-T MR scanner (Aera; Siemens Healthineers, Erlangen, Germany). To minimize motion artifacts, breath-holding techniques were employed. The imaging protocols consisted of the following: (a) axial, coronal, and sagittal half-Fourier acquisition single-shot turbo spin-echo with a field of view (FOV) of 420 × 80 mm, a 5 mm-thick section, a 20% gap, a matrix of 272 × 320, a repetition time (TR) of 1000 ms, a time to echo (TE) of 140 ms, and a scan duration of 50 s; (b) axial, coronal, and sagittal true fast imaging with steady-state precession with an FOV of 420 × 80 mm, a 5 mm-thick section, a 30% gap, a matrix of 234 × 384, a TR of 4.11 ms, a TE of 1.63 ms, and a scan duration of 48 s; (c) three-dimensional volumetric interpolated breath-hold examination with an FOV of 400 mm, a 5 mm-thick section, a 20% gap, a matrix of 180 × 320, and a scan duration of 8 s.

### 2.5. Imaging Analysis and Assessment of MRI Features of PAS Disorders

Blinded to the clinical and pathological findings after treatment, two experienced radiologists (with 9 and 15 years of experience, respectively) independently examined the MR images. Seven MRI features suggestive of PAS disorders were evaluated for the presence or absence of the following: T2 dark bands, placental heterogeneity, placental bulge, placental cervical protrusion sign, abnormal vascularization of the placental bed, focal exophytic mass, and myometrial thinning [7,9].

T2 dark bands: dark lines on T2-weighted images showing nodular or linear patterns from the uterus to the placenta (Figure 1a).Placental heterogeneity: uneven signal intensity observed inside the placenta, often due to repeated bleeding (Figure 1b).Placental bulge: lower uterine protrusion caused by the placenta, usually toward the bladder (Figure 1c).Placental cervical protrusion sign: placental tissue sticking to the cervical canal (Figure 1d).Abnormal vascularization of the placental bed: disturbed blood vessels in the placental bed affecting the uteroplacental connection (Figure 1e).Focal exophytic mass: placental tissue extending through the uterine wall (Figure 1f).Myometrial thinning: Thinning of the uterine muscle over the placenta, sometimes becoming nearly invisible (Figure 1g).

### 2.6. PABO Procedure

Before PABO, the patients were fully informed of the risks and complications. The procedure involved femoral artery puncture, sheath insertion, and placement of a 7 Fr occlusion balloon catheter. After successfully positioning the catheter, it was securely fastened to the skin. Subsequently, patients were moved from the interventional catheter room to the operating room for a CS. The CS procedure was performed under spinal anesthesia immediately following balloon placement. Following delivery and umbilical cord clamping, the occlusion balloons were inflated in accordance with the obstetrician’s instructions (Figure 2). After delivery, the balloons were deflated prior to skin closure. The radiologist removed the catheters once the patient’s vital signs stabilized, and compression was applied to the puncture sites.

### 2.7. Statistical Analysis

Continuous variables are expressed as means ± standard deviation or medians (range), and categorical variables as numbers (proportions, %). Statistical tests included Student’s *t*-tests, Mann–Whitney U-tests, and χ2 tests for comparing clinical and MRI features between the massive and non-massive bleeding groups. Statistical analyses determined the significant risk factors of MRI features for predicting adverse maternal outcomes during PABO and before CS. SPSS v. 22 software (IBM Corp., Armonk, NY, USA) was used for statistical analyses, with significance set at *p* < 0.05.

## 3. Results

In the present study, 40 patients with PAS and placenta previa underwent PABO during a CS. Four patients were excluded because of multiple pregnancies (*n* = 2) and planned hysterectomies (*n* = 2). The clinical characteristics and MRI features of the two groups are summarized in Table 1 and Table 2. Figure 3 shows a flowchart of the inclusion–exclusion process.

### 3.1. Background of the Participants

Among the 30 patients, 22 experienced massive bleeding (>2500 mL) and 13 underwent an emergency hysterectomy due to a refractory hemorrhage. Histopathological assessments in the cases of emergency hysterectomy revealed varying degrees of adherent placental severity: percreta (*n* = 3), increta (*n* = 3), and accreta (*n* = 16). Among the remaining 18 patients, there was minor bleeding (<2500 mL), and their uteri were successfully preserved. It is worth noting that patients in the nonmassive bleeding group resumed regular menstruation within a year, and no severe complications associated with balloon occlusion were noted. Importantly, there were no occurrences of maternal or neonatal deaths.

### 3.2. Comparison of Clinical Characteristics

Statistical analyses revealed significant differences between the nonmassive and massive bleeding groups across various parameters. Patients in the massive bleeding group had a significantly longer operative duration (213 ± 41.2 min vs. 100.4 ± 33.6 min; *p* < 0.001), a higher EBL (5822.1 ± 4223.4 mL vs. 1526.5 ± 842 mL; *p* < 0.001), an increased number of pRBC transfusions (10.7 ± 8.5 vs. 3.2 ± 4.1; *p* < 0.001), and a prolonged postoperative hospital stay (11.2 ± 4.2 days vs. 9.9 ± 3.2 days; *p* < 0.05).

### 3.3. MRI Features

A comparison of MRI features between the two groups revealed significant differences (10/18 (55.6%) vs. 20/22 (90.9%); *p* < 0.05) in the presence of T2 dark bands. However, no significant differences were observed in the other MRI features.

## 4. Discussion

Our investigation revealed that the T2 dark bands identified on MRI were significantly associated with massive hemorrhages and played a crucial role in predicting clinical outcomes in patients with PAS and placenta previa.

T2 dark band formation is believed to be caused by fibrin deposition [10]. In addition to their established roles in hemorrhage prediction, our findings highlight their implications for specific clinical interventions. The identified association with fibrin deposition not only strengthens their predictive value but also provides a biological rationale for their clinical significance.

In the realm of managing suspected PAS during a CS, the historical landscape shows a diverse array of surgical approaches, ranging from nonconservative methods, such as the B-Lynch suture, uterine artery embolization, uterine tamponade balloons, and ligation of the internal iliac artery, to the more invasive option of hysterectomy [11]. Amid this spectrum, PABO has emerged as a promising strategy to mitigate hemorrhages and reduce the demand for blood transfusions [12]. However, the challenge persists in discerning the specific cohort of patients who stand to gain the most from PABO.

Our previous report suggested that T2 dark bands and placental bulges observed on MRI may predict adverse maternal outcomes in patients with PAS and placenta previa undergoing prophylactic balloon occlusion of the internal iliac artery [7].

Upon occlusion of the internal iliac artery in a pregnant uterus, blood flow is promptly substituted by a comprehensive network of collateral arterial vessels [13]. PABO may effectively regulate the heightened blood flow linked to this condition. However, in cases of PAS with placenta previa, the extensive formation of collateral blood vessels and the blood flow related to T2 dark bands might not be manageable through PABO. Consequently, meticulous patient selection for prenatal PABO is crucial for personalized birth planning. Considering these factors, if T2 dark bands are present, opting for total hysterectomy from the outset, without proceeding with placental abruption, may be a preferable approach. The T2 dark bands, identified as significant MRI features in our study, have been previously recognized not only as predictors of a massive hemorrhage, but also as crucial factors in differentiating outcomes such as the EBL, pRBC transfusions, and plasma transfusions [14]. In our investigation, we consistently observed a higher frequency of T2 dark bands in the massive bleeding group than in the nonmassive bleeding group. The speculated mechanism involves fibrin deposition, leading to the formation of T2 dark bands, potentially resulting in a narrow intervillous space [15,16]. This narrowing of the intervillous space may prompt dilation or an increase in maternal vessels, including the spiral arteries and draining veins, in an attempt to enhance blood flow to the placenta [15,16]. However, the consequence of increased and/or dilated vessels is a potential exacerbation of hemorrhage during manual extraction of the placenta. This intricate interplay between T2 dark bands, fibrin deposition, and vascular changes underscores the multifaceted nature of the risk factors that contribute to adverse maternal outcomes in the context of PABO. The relationship between the T2 dark band and the extensive formation of collateral blood vessels and blood flow is still unclear and requires further investigation and pathological evaluation. In our study, the T2 dark band was the only risk factor. However, massive bleeding may occur even in the absence of a T2 dark band, and it may be necessary to study more cases, perform MRI evaluation, and consider the location, number, and volume of the involved areas.

In several management guidelines, a hysterectomy is recommended as the primary surgical approach in women with PAS [17]. In our study, we intended to preserve the uterus as much as possible through placental removal. This is because, taking into consideration the patient’s desire to ensure fertility, we tried to preserve as much of the uterus as possible with sufficient informed consent.

This study had some limitations. The speculated link to fibrin deposition requires further validation and the observational nature of our study precludes the establishment of direct causation. Notably, this study adopted a retrospective case–control design with a modest sample size. Only patients exhibiting equivocal ultrasound findings were directed for MRI, precluding the comparison of MRI results with pathological findings. It is crucial to recognize that our study was retrospective and conducted at a single center, introducing the possibility of selection bias. In addition, although two experienced radiologists independently evaluated the MR images, there may be some bias because they each consulted the other’s opinion. Furthermore, we could not ascertain an appropriate sample size because of the absence of comparable studies. Consequently, further prospective multicenter studies with larger sample sizes are needed to address these limitations and enhance the robustness of our findings.

In conclusion, although evaluating a limited number of cases, the identification of T2 dark bands on MRI is a valuable predictor of adverse maternal outcomes in patients with PAS and placenta previa undergoing PABO. Recognition of these findings in preoperative MRI may serve as a crucial indicator, prompting the need for proactive measures to predict potential a massive hemorrhage during a CS. Preparedness for effective management strategies is paramount for optimizing maternal outcomes, but further evaluation is required.

## Figures and Tables

**Figure 1 diagnostics-14-00333-f001:**
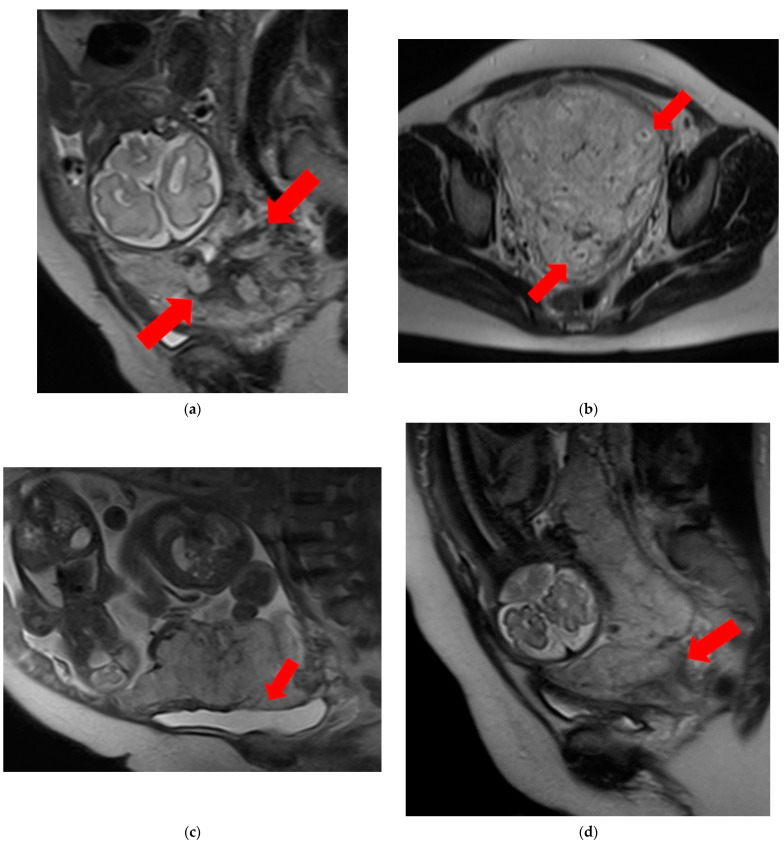
Examination of magnetic resonance imaging features in the spectrum of placenta accreta with placenta previa disorder reveals the following characteristics: (**a**) T2 dark bands, indicating areas of low signal intensity on T2-weighted images (arrows). (**b**) Placental heterogeneity, depicting varied signal intensity within the placenta due to recurrent hemorrhages or lacunae (arrows). (**c**) Placental bulge, characterized by the protrusion of the lower segment of the uterus caused by the mass effect of the placenta, typically towards the bladder (arrow). (**d**) Placental cervical protrusion, showcasing the extension of placental tissue into the cervical canal (arrow). (**e**) Abnormal vascularization of the placental bed, featuring noticeable vessels in the placental bed with disruption of the uteroplacental interface (arrows). (**f**) A focal exophytic mass, indicating the protrusion of placental tissue through the uterine wall and beyond (arrows). (**g**) Myometrial thinning, demonstrating a reduction in myometrial thickness over the placenta to <1 mm or even making it invisible (arrow).

**Figure 2 diagnostics-14-00333-f002:**
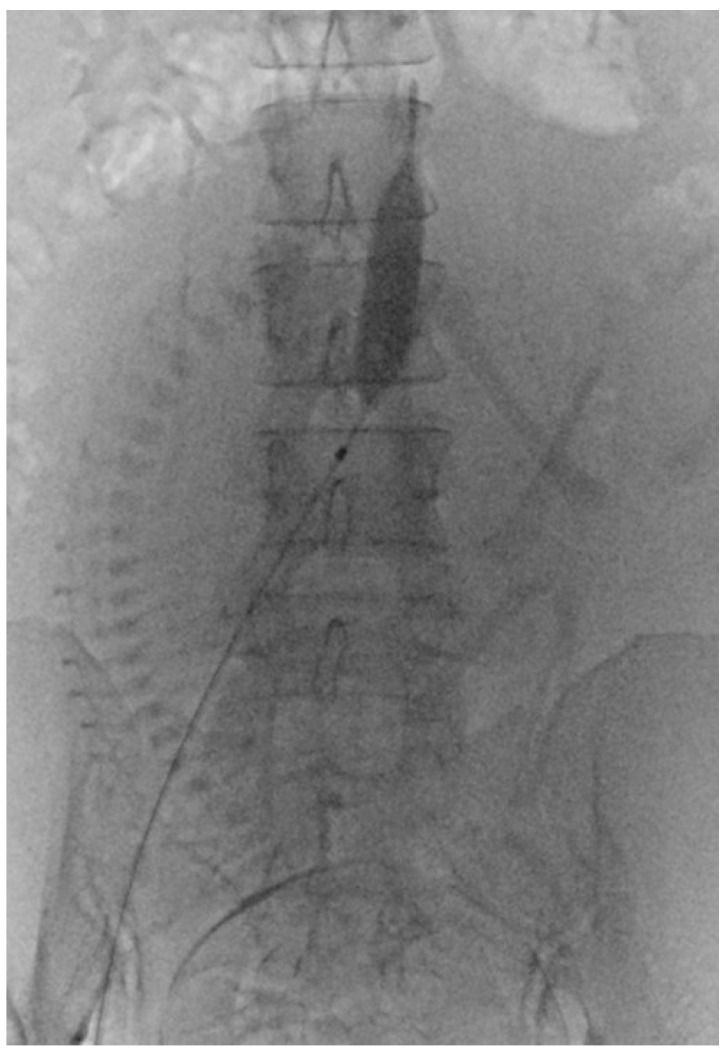
Prophylactic aortic balloon occlusion during a cesarean section for placenta accreta spectrum and placenta previa.

**Figure 3 diagnostics-14-00333-f003:**
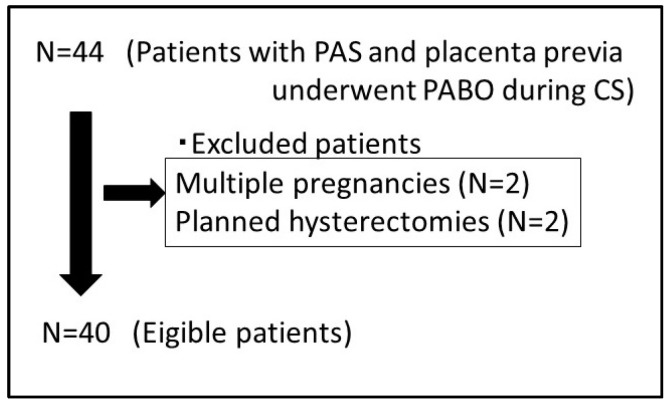
Flowchart of the inclusion–exclusion process.

**Table 1 diagnostics-14-00333-t001:** The clinical characteristics of the two groups.

	Nonmassive Bleeding Group	Massive Bleeding Group	*p*
N	18	22	
Age (years)	34.7 ± 4.2	36.2 ± 4.8	0.39
BMI (kg/m^2^)	26.8 ± 2.9	25.0 ± 2.7	0.72
Parity	1.5 ± 0.9	1.9 ± 0.5	0.29
Number of prior caesarean sections	1.2 ± 0.6	1.5 ± 0.6	0.48
Degree of placental adhesion			
Accreta	16	13	0.073
Increta	2	5	0.43
Percreta	0	4	0.11
Gestational age at the CS (days)	248 ± 13	251 ± 10	0.79
Gestational age at the MRI (days)	227 ± 10	129 ± 12	0.89
Foetal weight (g)	2482.6 ± 431.2	2641.2 ± 169.2	0.21
Operative duration (min)	100.4 ± 33.6	213 ± 41.2	<0.001
Estimated blood loss (mL)	1526.5 ± 842	5822.1 ± 4223.4	<0.001
Hysterectomy	0	13	<0.001
Number of pRBCs transfused	3.2 ± 4.1	10.7 ± 8.5	<0.001
Postoperative hospital stay (days)	9.9 ± 3.2	11.2 ± 4.2	0.47

BMI, body mass index; CS, cesarean section; MRI, magnetic resonance imaging; pRBCs, packed red blood cells.

**Table 2 diagnostics-14-00333-t002:** Magnetic resonance imaging features of the two groups.

	Nonmassive Bleeding Group	Massive Bleeding Group	*p*
T2 dark bands	10 (55.6%)	20 (90.9%)	0.025
Placental heterogeneity	8 (44.4%)	13 (59.1%)	0.53
Placental bulge	7 (38.9%)	13 (59.1%)	0.34
Placental cervical protrusion sign	7 (38.9%)	12 (54.5%)	0.36
Abnormal vascularization of the placental bed	3 (16.7%)	9 (40.9%)	0.17
Focal exophytic mass	8 (44.4%)	11 (50.0%)	0.76
Myometrial thinning	8 (44.4%)	9 (40.9%)	1.00

## Data Availability

The data presented in this study are available on request from the corresponding author.

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
