# Peer review of "MRI-Based Risk Factors for Adverse Maternal Outcomes in Prophylactic Aortic Balloon Occlusion for Placenta Accreta Spectrum and Placenta Previa"

_diagnostics, 2024, doi:10.3390/diagnostics14030333_

Round 1

Reviewer 1 Report

Comments and Suggestions for Authors

This study was about to investigate MRI findings related to outcomes of using new vascular control technique "prophylactic aortic balloon occlusion"  in PAS

The scientific background sounds interesting but there are several flaws that raise some concerns regarding a conduct of this paper.

1- Inconsistency on number of patient population throughout this paper. It is confused whether the research included 40 or 30 patients,  for example;

    Study design page 2 line73= we conducted of 40 consecutive patients.....

    Results page 6 line 170=  in this present study, 30 patients with PAS and placenta previa underwent PABO during CS. Four patients were excluded because .....

     Table 1= N for nonmassive bleeding group=18 and massive bleeding group= 22; so the final population =40?

    Background of the participants page7 line 180= amoung the 30 patients, 22 experienced massive bleeding.......

2. Several flaws on labelling or image representations of signs on Figure 1 a, c,f,g.

Please carefully review descriptions for each MRI findings according to Jha P, et al. Society of Abdominal Radiology (SAR) and European Society of Urogenital Radiology (ESUR) joint consensus statement for MR imaging of placenta accreta spectrum disorders. Eur Radiol. 2020 May;30(5):2604-2615 (mentioned as reference No 9 this work). 

The most concern is the figure 1A for T2 dark bands which is the key features of this paper.  Given the description of T2 dark bands  are irregular bands which typically extend from the maternal surface. Figure 1A was not the T2 dark bands but intraplacental vessels. Then this is liable to a concern of error on MRI features interpretation.

3. Method of patients selection

Authors have clearly stated that (page 2 line 78) Cs were scheduled and the decision to perform hysterectomy was made based on intraoperative findings. In all cases, the objective was to preserve the uterus as much as possible through placental removal, uterine reconstruction and the use of PABO.  

As per several management guidelines,  hysterectomy is recommended as the primary surgical approach in women with PAS and options eg. partially resective technique or placental conservation are recommended as alternative to hysterectomy. 

Ref. Capannolo G,  et al.. Placenta accreta spectrum disorders clinical practice guidelines: A systematic review. J Obstet Gynaecol Res. 2023 May;49(5):1313-1321. 

In this study, the patients were diagnosed PAS included degree of invasion preoperatively so that please clarify more about why the patients with known myoinvasive placenta (ie, increta and percreta) did not have planned hysterectomy.

Author Response

Reviewer 1

This study was about to investigate MRI findings related to outcomes of using new vascular control technique "prophylactic aortic balloon occlusion"  in PAS The scientific background sounds interesting but there are several flaws that raise some concerns regarding a conduct of this paper.

1- Inconsistency on number of patient population throughout this paper. It is confused whether the research included 40 or 30 patients,  for example;

    Study design page 2 line73= we conducted of 40 consecutive patients.....

    Results page 6 line 170=  in this present study, 30 patients with PAS and placenta previa underwent PABO during CS. Four patients were excluded because .....

     Table 1= N for nonmassive bleeding group=18 and massive bleeding group= 22; so the final population =40?

    Background of the participants page7 line 180= amoung the 30 patients, 22 experienced massive bleeding.......

In the present study, 40 patients with PAS and placenta previa underwent PABO during CS.

  1. Several flaws on labelling or image representations of signs on Figure 1 a, c,f,g.

Please carefully review descriptions for each MRI findings according to Jha P, et al. Society of Abdominal Radiology (SAR) and European Society of Urogenital Radiology (ESUR) joint consensus statement for MR imaging of placenta accreta spectrum disorders. Eur Radiol. 2020 May;30(5):2604-2615 (mentioned as reference No 9 this work). 

The most concern is the figure 1A for T2 dark bands which is the key features of this paper.  Given the description of T2 dark bands  are irregular bands which typically extend from the maternal surface. Figure 1A was not the T2 dark bands but intraplacental vessels. Then this is liable to a concern of error on MRI features interpretation.

Figure 1 a was changed.

  1. Method of patients selection

Authors have clearly stated that (page 2 line 78) Cs were scheduled and the decision to perform hysterectomy was made based on intraoperative findings. In all cases, the objective was to preserve the uterus as much as possible through placental removal, uterine reconstruction and the use of PABO.  

As per several management guidelines,  hysterectomy is recommended as the primary surgical approach in women with PAS and options eg. partially resective technique or placental conservation are recommended as alternative to hysterectomy. 

Ref. Capannolo G,  et al.. Placenta accreta spectrum disorders clinical practice guidelines: A systematic review. J Obstet Gynaecol Res. 2023 May;49(5):1313-1321. 

In this study, the patients were diagnosed PAS included degree of invasion preoperatively so that please clarify more about why the patients with known myoinvasive placenta (ie, increta and percreta) did not have planned hysterectomy.

In several management guidelines, hysterectomy is recommended as the primary surgical approach in women with PAS [17]. In our study, we intended to preserve the uterus as much as possible through placental removal. This is because, taking into consideration the patient's desire to ensure fertility, we tried to preserve as much of the uterus as possible with sufficient informed consent.

Reviewer 2 Report

Comments and Suggestions for Authors

1. Interesting cohort but more understanding and detail required.

2. line 170 : these excluded cases should just be included in separate Table for limited information and comment.

3. line 189-193: this is expected in the group once bleeding starts does not help re prediction of bleeding.

4. line 212: if T2 dark bands are the only MRI feature then more needs MRI features need to be done re location , number , volume of area involved etc as non 10/18 has dark bands vs bleeding 20/22 but need to know before not after.

5. line 219-221: more explanation why the occlusion does not work as very important next comment.

6. More discussion as you indicate a goal is to preserve the uterus but 20/22 had dark bands and 13/22 has a hyst (while 10/18 had dark bands but no hyst) so decision re leave placenta attached and do a hyst or separate and see the result requires more discussion and consideration. Better thought re hyst or no hyst ? parity; other health issues; personal choice, as doing a hyst that is not required is a problem unless informed consent to the best of the prediction is done.

Comments on the Quality of English Language

Reasonably well written.

Author Response

Reviewer 2

  1. Interesting cohort but more understanding and detail required.

Thank you for your sugggestions

  1. line 170 : these excluded cases should just be included in separate Table for limited information and comment.

We added a Figre3.

  1. line 189-193: this is expected in the group once bleeding starts does not help re prediction of bleeding.

Your opinion is valid, but we included it to show the patient's background.

  1. line 212: if T2 dark bands are the only MRI feature then more needs MRI features need to be done re location , number , volume of area involved etc as non 10/18 has dark bands vs bleeding 20/22 but need to know before not after.

We added the following sentence.

In our study, T2 dark band was the only risk factor. However, massive bleeding may occur even in the absence of a T2 dark band cases, and it may be necessary to collect more cases and perform MRI evaluation, and to consider the location, number, and volume of the involved areas.

  1. line 219-221: more explanation why the occlusion does not work as very important next comment.

The relationship between the T2 dark band and the extensive formation of collateral blood vessels and the blood flow is still unclear and requires further investigation and pathological evaluation.

  1. More discussion as you indicate a goal is to preserve the uterus but 20/22 had dark bands and 13/22 has a hyst (while 10/18 had dark bands but no hyst) so decision re leave placenta attached and do a hyst or separate and see the result requires more discussion and consideration. Better thought re hyst or no hyst ? parity; other health issues; personal choice, as doing a hyst that is not required is a problem unless informed consent to the best of the prediction is done.

In several management guidelines, hysterectomy is recommended as the primary surgical approach in women with PAS [17]. In our study, we intended to preserve the uterus as much as possible through placental removal. This is because, taking into consideration the patient's desire to ensure fertility, we tried to preserve as much of the uterus as possible with sufficient informed consent.

Reviewer 3 Report

Comments and Suggestions for Authors

An interesting study about a quite specific problem. As a result methodology and the statistics should be more solid. 

1. Please add a flowchart about the inclusion-exclusion process

2. The authors said: "Blinded to the clinical and pathological findings after treatment, two experienced ra- 117 diologists (with 9 and 15 years of experience, respectively) independently examined the 118 MR images." however they did not clarify how the final desicion was made: with concensus? by a third researcher? did they study interobserver reliability? if not, it should be stated as a limitation

3. Regression analysis is an important test to define the effect and the power of the potential risk factors. The authors should use regression analysis at least for T2 dark bands, but preferably to all parameters. 

4. Results 3.2 and 3.3: These information should be given with percentages and numeric data. I guess these paragraphs refer to table 2, the authors should mention this reference in the text. 

5. Table 2 should contain relevant percentages. 

6. Conclusion is too conscise for a study of 30 patients. Please prefer more soft expressions and be less precise. 

Author Response

Reviewer 3

An interesting study about a quite specific problem. As a result methodology and the statistics should be more solid. 

  1. Please add a flowchart about the inclusion-exclusion process

We added Figure 3. Flowchart about the inclusion-exclusion process.

  1. The authors said: "Blinded to the clinical and pathological findings after treatment, two experienced ra- 117 diologists (with 9 and 15 years of experience, respectively) independently examined the 118 MR images." however they did not clarify how the final desicion was made: with concensus? by a third researcher? did they study interobserver reliability? if not, it should be stated as a limitation

In addition, although two experienced radiologists independently evaluated the MR images, there may be some bias because they consulted each other's opinions.

  1. Regression analysis is an important test to define the effect and the power of the potential risk factors. The authors should use regression analysis at least for T2 dark bands, but preferably to all parameters. 

Your suggestion is the best, but because the number of cases is small and the parameters are not continuous variables, it is not possible to conduct a sufficient statistical study. Therefore, it is necessary to accumulate more cases.

  1. Results 3.2 and 3.3: These information should be given with percentages and numeric data. I guess these paragraphs refer to table 2, the authors should mention this reference in the text. 

3.2. Comparison of Clinical Characteristics

Statistical analyses revealed significant differences between the nonmassive and massive bleeding groups across various parameters. Patients in the massive bleeding group had a significantly longer operative duration (213 ± 41.2min vs 100.4 ± 33.6min; p <0.001), higher EBL (5822.1 ± 4223.4ml vs 1526.5 ± 842ml; p <0.001), increased number of pRBC transfusion (10.7 ± 8.5 vs 3.2 ± 4.1; p <0.001), and prolonged postoperative hospital stay (11.2 ± 4.2days vs 9.9 ± 3.2days; p <0.05).

3.3. MRI Features

A comparison of MRI features between the two groups revealed significant differences (10/18 (55.6%) vs 20/22 (90.9%); p <0.05) in the presence of T2 dark bands. However, no significant differences were observed in the other MRI features.

  1. Table 2 should contain relevant percentages. 

Tbale2 was changed

  1. Conclusion is too conscise for a study of 30 patients. Please prefer more soft expressions and be less precise. 

In conclusion, although evaluating a limited number of cases, the identification of T2 dark bands on MRI is a valuable predictor of adverse maternal outcomes in patients with PAS and placenta previa undergoing PABO. Recognition of these findings on preoperative MRI may serve as a crucial indicator, prompting the need for proactive measures to predict potential massive hemorrhage during CS. Preparedness for effective management strategies is paramount for optimizing maternal outcomes, but further evaluation is required.

Round 2

Reviewer 1 Report

Comments and Suggestions for Authors

Still concern over severe flaws on context, images and about patient selection in this research.

Quality of paper regarding scientific background is inadequate to get published in the high impact factor paper.

Reviewer 2 Report

Comments and Suggestions for Authors

My concerns and questions were answered , thankyou.

Comments on the Quality of English Language

Minor issues.

Reviewer 3 Report

Comments and Suggestions for Authors

The recommendations were mostly performed thank you